# A systematic review and meta-analysis of knowledge, attitudes, and practices toward anthrax prevention and control in Ethiopia: Implication for a one health policy

Ayenew Takele Alemu[1]*, Birtuedil Yibeltal Beyene[2], Gashaw Molla Beza[3], Friehiwot Molla[4], Wolde Melese Ayele[1], Gashaw Melkie Bayeh[2], Mahider Awoke Belay[1], Kindie Bayih Geremew[5], Abathun Temesgen[4], Mekuanint Taddele Tessema[1], Melaku Laikemariam[6], Almaw Genet Yeshiwas[2], Atirsaw Assefa Melikamu[1]

**1** Department of Public Health, College of Medicine and Health Sciences, Injibara University, Injibara, Ethiopia, **2** Department of Environmental Health, College of Medicine and Health Sciences, Injibara University, Injibara, Ethiopia, **3** Department of Veterinary Laboratory, College of Agriculture, Injibara University, Injibara, Ethiopia, **4** Department of Public Health, College of Medicine and Health Sciences, Debre Markos University, Debre Markos, Ethiopia, **5** Department of Nursing, College of Medicine and Health Sciences, Injibara University, Injibara, Ethiopia, **6** Department of Midwifery, College of Medicine and Health Sciences, Injibara University, Injibara, Ethiopia

* ayetake21@gmail.com

## Abstract

### Background

Anthrax is one of the tropical diseases that are often overlooked... Anthrax's burden extends beyond its effects on health; it also has an economic cost. Implementing One Health policies requires knowledge, attitude, and practice (KAP) about anthrax prevention and control. However, there is no nationally aggregated evidence in Ethiopia. Therefore, this review was done to generate evidence and offer suggestions for incorporating KAP findings into One Health strategy to prevent and control anthrax in Ethiopia.

### Methods

The protocol was registered in a PROSPERO with a reference CRD420251141478 and Preferred Reporting Items for Systematic Review and Meta-analysis (PRISMA) guidelines were adhered. The electronic databases: Pub-Med, Scopus, Web of Science, CAB abstracts, AGRICOLA, and Google-Scholar were searched to retrieve the included studies using key search terms with database specific search strings. Studies that reported KAP outcomes in Ethiopian and were published only in English language irrespective of publication time were included in this review. The random-effects model was used to pool effect sizes using STATA 17 software. Sub-group and meta-regression analyses were performed to explore potential sources of

**Data availability statement:** All the required data are available in the manuscript.

**Funding:** The author(s) received no specific funding for this work.

**Competing interests:** The authors have declared that no competing interests exist.

heterogeneity, while sensitivity analysis was performed to assess the robustness of the pooled estimates. Forest plots were used to display the results.

## Results

This study included 17 articles with 8,369 participants that met the inclusion criteria. Our synthesis found that consuming raw meat, backyard slaughtering, improperly disposing of carcasses, and sharing a home with animals are common risk factors for anthrax infection in Ethiopia. The pooled knowledge, attitude, and practice levels of anthrax prevention and control were 51.25% (95% CI: 43.93, 58.58; $I^2 = 98.01\%$), 59.26% (95% CI: 50.43%, 68.08%; $I^2 = 98.33\%$), and 50.62% (95% CI: 42.95, 58.29; $I^2 = 97.93\%$), respectively.

## Conclusions

Half of the communities in Ethiopia remain with suboptimal knowledge, attitudes, and practices for anthrax prevention and control. This could lead to ineffective outbreak management, delayed reporting, and continuous transmission of anthrax to humans and animals, particularly in endemic areas. As a result, from a One Health perspective, an integrated multisectoral intervention is urgently required to promote collaboration among the human, animal, and environmental health sectors.

## Author summary

### Why was this study done?

Anthrax is a common animal-to-human transmissible disease that is often overlooked. However, Ethiopian communities' knowledge, attitude, and practice have not been evidenced nationally. The purpose of this study was to give the first national estimates of community knowledge, attitude, and practice related to anthrax prevention and control in Ethiopia, as well as insights into a One Health policy.

### What did the researchers do and find?

We pooled and analysed the findings of all relevant studies on KAP outcomes related to anthrax in Ethiopia. The key findings were:

- Raw meat consumption, animal slaughter, and cohabitation with animals have been identified as common anthrax risk factors in Ethiopia.

- The combined percentages of good knowledge, favourable attitudes, and safe practices for anthrax prevention and control are 51.25%, 59.26%, and 50.26%, respectively.

- Subgroup analyses revealed regional, population, and study period related variations in KAP outcomes.

- Over the past ten years, practice outcome has minimally increased, while knowledge and attitude outcomes have stayed the same, showing no temporal trends.

## What do these findings mean?

Anthrax prevention and control KAP outcomes in Ethiopia are suboptimal. This could result in on-going animal-to-human transmission, underreported cases, and an inefficient outbreak response. The adoption of One Health, public health education, and veterinary services should be prioritized in order to improve anthrax prevention and control in Ethiopia.

## Introduction

Anthrax, a widespread zoonotic disease, is caused by *Bacillus anthracis* that may survive for a long time in soil and animal products in light of its highly resistant spores [1,2]. Anthrax infects all mammals; however, the most commonly affected animals are cattle, goats, sheep, and horse [3]. Grazing on contaminated pastures is a primary means of acquiring infection for herbivores through ingestion or inhalation of spores [4]. Humans can contract the disease by eating raw or undercooked meat, coming into contact with the carcasses of infected animals, or coming into contact with animal products contaminated with spores [5]. The 95% of all cases of anthrax in humans have cutaneous illness, making it the most prevalent form of the anthrax [6]. When left untreated, human cutaneous anthrax causes 5–20% of deaths; however, when detected and treated early, the mortality rate is low; for instance, in Ghana, untreated human cutaneous anthrax was responsible for 35% of case-fatality rates between 2005 and 2016 [7,8]. On the other hand, human anthrax can also be inhaled or gastrointestinal, and when treatment is delayed, it can result in a high case-fatality rate of 25–75% [3,9].

Anthrax is one of the tropical diseases that are often overlooked. An estimated 1.83 million individuals worldwide live in areas at risk of anthrax. The World Health Organisation (WHO) estimates that 20,000–100,000 cases of human anthrax occur each year worldwide [10,11]. A meta-analysis study evidenced that a global burden of livestock anthrax accounted for 28% with the highest prevalence in African countries (29%) [12]. In Ethiopia, anthrax is endemic and often occurs between May and June, which is referred to as the "*anthrax season*" affecting both humans and animals [13]. According to a study done in Awi zone, Ethiopia between 2011 and 2020, there were 1,262 incidences of animal anthrax and 324 fatalities [1]. From the year 2008–2012, 2,680 cases of human anthrax were reported to public health facilities in Northern Ethiopia [14]. Despite being an endemic zoonotic disease in Ethiopia, the public health burden of anthrax is underreported, and there is little multi-sectoral integration to control it. Recent research and national reports indicate that anthrax is still being spread in a number of areas; as a result, there is a pressing need for community awareness, diagnostic capabilities, and surveillance [15].

The burden of Anthrax extends beyond its negative effects on health; it also has an economic cost. Unexpected animal deaths due to anthrax epidemics frequently result in financial losses that can increase a household's monthly income. Consequently, families become suffer to food insecurity [16]. The expenses incurred for epidemic response and investigation also contribute to economic hardship. According to reports, a single multi-woreda response to the anthrax outbreak cost 102,232 US dollars for travel, investigation, vaccination programs, and carcass disposal [17]. Ethiopia and other developing nations that depend on livestock production are particularly hard affected by the prevalence of animal illnesses like anthrax. A recent review demonstrated that a livestock illness has major macroeconomic impacts, with improved animal health potentially increasing GDP by up to 3.6% [18].

Anthrax cases are more likely to go unreported and animals may be slaughtered or their meat eaten when members of the community are unaware of the disease. Therefore, human cases won't be noticeable till exposure has taken place. Inadequate knowledge delays detection and raises exposure risk. Even when people are aware of the dangers of anthrax, negative attitudes may decrease the number of cattle receiving the vaccine and increase the use of risky carcass handling

practices, eating meat from suddenly died animals, and backyard slaughtering close to homes [19]. These behaviours are direct routes for persistent spores to infect humans and contaminate the environment [20].

A one health approach is crucial to preventing and controlling the outbreak of anthrax [5]. Ethiopia has launched the One Health strategy, which integrates the human, animal, and environmental sectors, to prevent and control zoonotic diseases. However, there is little evidence showing that One Health implementation strategies reach community members in terms of information, attitude, and behaviour [21].

Implementing One Health policies requires an understanding of the knowledge, attitude, and practices (KAP) of community members at risk for anthrax, livestock owners, and human and animal health professionals. Evidence on community KAP regarding anthrax in Ethiopia is fragmented and inconsistently reported, despite numerous epidemic investigations and facility- and community-based studies were conducted. On this topic, no nationally pooled evidence exists. Therefore, a systematic review and meta-analysis was conducted to close the evidence gap by examining the initial estimates of knowledge, attitudes, and practices (KAP) related to anthrax and to offer useful suggestions for incorporating KAP results into One Health policy for anthrax prevention and control initiatives.

## Methods and materials

### Protocol registration and guidelines

We registered the protocol in the PROSPERO database of systematic reviews with registration identification reference CRD420251141478 after performing a preliminary search to demonstrate the viability of performing a systematic review and meta-analysis on this topic. The Preferred Reporting Items for Systematic Reviews and Meta-analysis (PRISMA) 2020 standards for observational studies were followed in doing this systematic review and meta-analysis [22].

### Search strategies and review questions

A comprehensive search of numerous electronic databases, including Pub-Med, Scopus, Web of Science, CAB abstracts, and AGRICOLA, was done to find relevant studies. We also looked for grey literature using other search engines, such as Google Scholars and the repositories of Ethiopian Universities. To add the studies we missed when searching electronic databases, we also performed a snowball search using the reference lists of the articles that were found. Publication year was not restricted during the literature search to ensure comprehensive identification of all relevant KAP studies on anthrax prevention and control in Ethiopia.According to the Joanna Bridges Institute's (JBI) guidelines for systematic reviews of prevalence studies, we developed the research questions using the condition, context, population, and outcome of interest (CoCoPop) framework [23]. In this study, population (Po) refers to community members, health workers, veterinarians, environmental health professionals, and livestock owners or animal handlers who are at risk of anthrax exposure. Condition (Co) refers to knowledge, attitude, and practices (KAP) related to anthrax prevention and control; and context (Co) refers to low- and middle-income countries, specifically Ethiopia. Thus, "what are the prevalence of good knowledge, favourable attitude, and safe practice regarding anthrax prevention and control, and what implications do these have for a One Health approach in Ethiopia?" was the structured research question of this systematic review and meta-analysis. This review was also aimed to test a hypothesis stated as "the pooled levels of knowledge, favourable attitude, and appropriate practices regarding anthrax prevention and control in Ethiopia vary significantly across populations, study years, and regions".

Using the Medical Subject Heading (MeSH) terms and a combination of keywords such as "anthrax," "Bacillus anthracis," "Knowledge," "attitude," "practice," "KAP," "awareness," "perception," "believe,"," "behavior," "behaviour," "one health," and "Ethiopia," all relevant studies were accessed. Advanced search using database specific search strings that were developed by Boolean operators "AND" and "OR" was carried out (S2 File). The literatures' searching was conducted from August to September, with last searching date 15 September 2025. The articles were downloaded, screened, and cited using EndNote version 20 references management software.

## Eligibility criteria

We established the eligibility criteria based on CoCoPop framework. We included all cross sectional/survey studies that reported quantitative and qualitative knowledge, attitude, and practice (KAP) outcomes on human and animal anthrax among community members, livestock owners, health workers, veterinarians, or environmental health professionals; studies conducted in any Ethiopian setting;; and studies published exclusively in English. Both peer-reviewed (published) and grey studies were considered to be included without regard to publication or study years' restrictions even though grey literatures couldn't be accessed, assessed, and included. However, this systematic review and meta-analysis excluded reviews or case reports, studies that were solely qualitative and focused on clinical treatment outcomes, laboratory diagnosis, or pathophysiology. Two authors (ATA and MAB) independently evaluated the qualifying criteria, and any discrepancies were settled by careful discussion and consensus.

## Risk of bias (Quality) assessment

Two authors (GM & AGY) independently evaluated the methodological quality of the included papers using the Joanna Bridge Institute (JBI) Critical Appraisal Checklist for prevalence studies, [24]. Adequacy of sample size, sampling method and recruitment strategy, statistical analysis and handling of missing data, validity of outcome measurement, representativeness of respondents in compliance with the checklist, and appropriateness of sampling procedure were all critically examined in the risk of bias assessment. Each eligible study was assessed using an adapted eight question items checklist that were scored as "Yes," "No," "Unclear," or "Not applicable." Our analysis was limited to studies with five (>60%) overall quality scores for "Yes" answers. Disagreements between authors were settled by conversation, and a third author (AAM) was designated to assist with the risk of bias assessment.

## Outcome measurements and definitions

Pooled prevalence of good knowledge, favorable attitude, and safe practice towards anthrax prevention and control were primary outcomes of this systematic review and meta-analysis. Heterogeneous KAP definitions and scoring systems were harmonized using a predefined algorithm that standardized outcomes into binary categories prior to meta-analysis. Our study interest was to pool effect sizes for good knowledge, favorable attitude, and appropriate practice. Proportions of study participants who had adequate or good knowledge of anthrax about its cause, symptoms, mode of transmissions, prevention, and treatment were taken to measure knowledge outcomes. Attitude outcome was pooled by taking proportions of study participants with favorable responses towards anthrax prevention and control strategies. Regarding practice outcome, we took proportions of study participants who practiced the recommended prevention and control measures. Narratively synthesized evidences on the reported risks of acquiring anthrax among humans and animals, and reported gaps of One Health approach were secondary outcomes of this systematic review.

## Data extraction

The included studies were coded and relevant data were extracted from each study consistently using a standardized data extraction format developed in Microsoft Excel Spread sheets for window. The elements of data extracted consisted of: first author name, publication year, data collection (study) year, study region (settings), study design, population type, sample size, sampling technique, data collection technique, local name to anthrax, gender distribution of participants, response rate, heard of anthrax (%), knows animal-human transmission (%), proportions of knowledge, attitude, and practice, reported risk for anthrax infection, reported preventive and control strategies, and reported one health approach. Two authors (ATA & GMB) independently extracted the data. A third author (AT) facilitated the process and disagreements between authors were addressed through dialoguing and discussion.

## Data management and analysis

For further statistical analysis, the extracted data were imported into the STATA 17 program. We generated and presented narratively study characteristics data, reported anthrax infection risks, reported anthrax prevention and control strategies, and One Health report and its gaps using a systematic review approach. To determine the pooled effect sizes of KAP outcomes, we performed a meta-analysis using a random effects model with generated 95% CIs [25].

Cochran's Chi-square (Q-test) and inverse-variance ($I^2$) statistical tests were used to assess the degree of heterogeneity among the included studies at p-values of less than 0.05 levels of significance. Tables and forest plots were used to present the findings [26,27]. Using study regions, publication years, and population types as categories of the included studies, sub-group analyses were performed to demonstrate variability of effect sizes for KAP outcomes. To find possible sources of heterogeneity among the included studies, sample size and publication year were used as covariates in a meta-regression analysis. By examining funnel plots and Egger's test at a p-value of less than 0.05, the presence of publication bias in calculating effect sizes of KAP outcomes was evaluated. Additionally, we conducted sensitivity analysis by leaving out every study that was included in order to look at how each study affected the overall results [28,29].

## Results

Searching sources yielded 1,184 records in total; 98 articles were eliminated as duplicates. After reviewing titles and abstracts, 1,040 articles were found irrelevant and were subsequently removed. finally this review included 17 studies following full-text eligibility evaluation (Fig 1).

## Characteristics of the included studies

Out of 17 included studies, nine studies [30–38] were conducted in Oromia, five studies [39–43] in Amhara, one study [44] in Tigray, one study [45] in Southern nations nationalities and people (SNNP), and one study [46] in Afar and Somalia regions, Ethiopia. Even though we considered both published and unpublished articles, all of the included studies

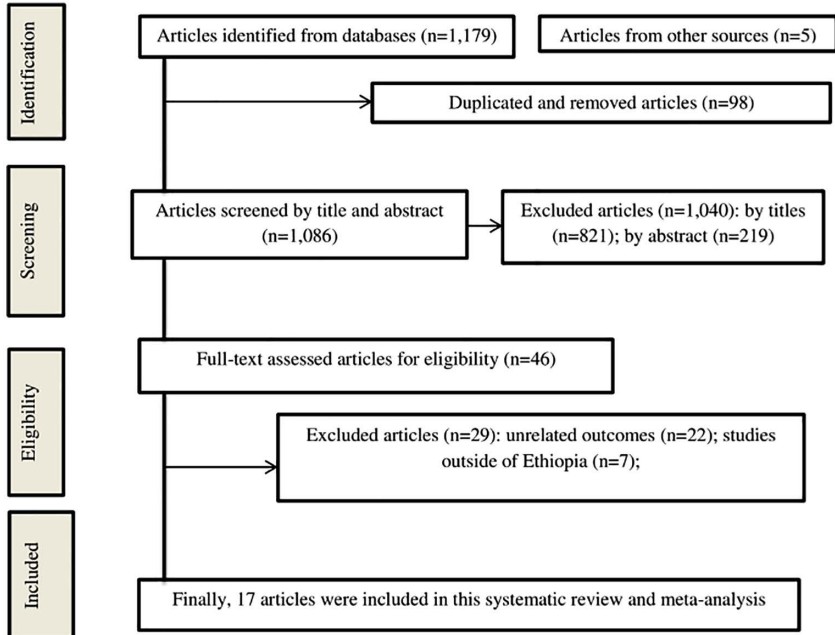

**Fig 1. PRISMA flow chart for selecting study articles.**

were published; grey literatures couldn't be accessed; therefore, unpublished studies were not assessed or included. Sixteen studies were cross-sectional in study design and one study [44] was cross-sectional triangulated with qualitative method. Publication and data collection years in the included studies range from 2017-2025 and 2016–2024, respectively. Regarding outcome measurement, all studies reported knowledge outcome, but three studies [35,39,46] and two studies [35,46] didn't report attitude and practice outcomes, respectively. Study populations were livestock owners in six studies [31,32,39,40,42,43], general community in eight studies [30,33–37,44,45], farmers in one study [41], pastoralists in one study [46], and professionals in one study [38]. Ten studies assessed KAP outcomes only on anthrax prevention and controlling [31,32,34,38,40–45] and the remaining seven studies [30,33,35–37,39,46] assessed KAP on zoonotic diseases including anthrax prevention and control.

A total of 8,369 study participants were involved in primary studies from which male participants accounted for 69.4%. Simple random sampling was employed in nine studies [30–33,35,36,39,40,46], systematic sampling in five studies [34,41,42,44,45], and purposive sampling in three studies [37,38,43]. Fifteen studies collected data using face-to-face interviews and the rest two studies [39,40] used self-administered interview. The reported response rates range from 90.91%-100% (Table 1).

## Anthrax related characteristics

All studies that were included noted whether or not the participants had information about anthrax. The proportions of participants who heard on anthrax range from the minimum 45.1% [36] to the maximum 99.9% [32]. Except two studies [35,41], all included studies reported proportions of participants who knew animal-to-human transmission of anthrax. The least and maximum proportions of participants awareness on transmission accounted for 21.3% [44] and 97.7% [32], respectively. Our systematic review identified that anthrax disease has locally adapted nomenclatures. It has been known as "*Abba Sanga*" [31,32] and "*Qoorra*" [34] in Oromia, "*Megerem*" ("*Migli Chiwa*" for human anthrax and "*Zigzag*" for animal anthrax) [44] in Tigray, "*Duluwa*" [45] in SNNP, and "*Qurba*" [43] in Amhara regions.

Table 1.  Characteristics of the included studies in a systematic review and meta-analysis of knowledge, attitude, and practice on anthrax prevention and control in Ethiopia, 2025.

| First author | Publication year | Study year | Study region | Study design | Study population | Sample size | Proportion of knowledge | Proportion of attitude | Proportion of practice | Quality score |
|---|---|---|---|---|---|---|---|---|---|---|
| Abebaw et al | 2025 | 2024 | Amhara | CS | Livestock owners | 771 | 43.5% | didn't report | 58.6% | 7/8 |
| Seid et al | 2019 | 2018 | Amhara | CS | Livestock owners | 800 | 58% | 53% | 25.4% | 6/8 |
| Dula et al | 2024 | 2022 | Oromia | CS | General | 392 | 56.4% | 80.8% | 55.8% | 7/8 |
| Mesfin et al | 2021 | 2020 | Amhara | CS | Farmers | 1295 | 58% | 48% | 35.8% | 8/8 |
| Tschopp et al | 2024 | 2019 | Afar/Somalia | CS | Pastoralists | 560 | 25.46% | didn't report | didn't report | 7/8 |
| Kabeta et al | 2018 | 2017 | Oromia | CS | Livestock owners | 201 | 67.2% | 78.6% | 71.6% | 6/8 |
| Adugna et al | 2025 | 2024 | Oromia | CS | Livestock owners | 384 | 59.07% | 77.43% | 56.3% | 7/8 |
| Teklu et al | 2018 | 2016 | Oromia | CS | General | 384 | 33.77% | 47.63% | 54.82% | 7/8 |
| Romha et al | 2020 | 2020 | Tigray | Mixed | General | 800 | 35.96% | 26.34% | 33.99% | 8/8 |
| Awraris et al | 2024 | 2022 | Amhara | CS | Livestock owners | 190 | 44.74% | 77.4% | 74.6% | 7/8 |
| Waraksa et al | 2024 | 2023 | Oromia | CS | General | 384 | 51.74% | 34.22% | 25.94% | 6/8 |
| Kerorsa eta al | 2019 | 2017 | Oromia | CS | General | 440 | 79.55% | didn't report | didn't report | 6/8 |
| Asha et al | 2021 | 2021 | SNNP | CS | General | 384 | 67.2% | 57.2% | 42.45% | 6/8 |
| Abrahim et al | 2024 | 2021 | Oromia | CS | General | 350 | 45.1% | 70.28% | 56.57% | 7/8 |
| Bsrat et al | 2017 | 2016 | Oromia | CS | General | 384 | 49.5% | 60.33% | 47.96% | 8/8 |
| Bezie et al | 2024 | 2022 | Amhara | CS | Livestock owners | 250 | 26.56% | 55.44% | 59.6% | 7/8 |
| Getahun et al | 2023 | 2022 | Oromia | CS | Health workers | 400 | 69.55% | 63.6% | 61.8% | 8/8 |

## Anthrax risk factors, prevention strategies and one health related reports

Consumption of raw meat [30,33–35,39,45,46], backyard slaughtering [37,39,45,46], poor disposal system for carcasses [30,33,34,45], traditional (cultural) beliefs [32,33,44], free grazing system [34], and living with animals in the same house and handling dead animal bodies without using protective equipment [37,45] are identified risk factors of acquiring human and animal anthrax infection., there are also preventive strategies that were indicated and recommended in the primary studies. These include: awareness creation and public education [30,32–35,37,39,41–43,45], keeping animal health through vaccination [34,39,43,45,46], proper carcasses disposal system using protective equipment [30,39,45], avoiding backyard slaughtering and setting regulatory system [34,39], early detection and reporting of outbreaks through strengthened surveillance [33,42,45,46], and mass-media communication [37,45]. One Health as a strategic approach for preventing and controlling anthrax outbreaks was indicated only in eight [33,34,36,38,39,42,45,46] of the included studies. Absence of clear protocol, limited resources, poor collaboration among animal and human health sectors, and poor communication were reported One Health gaps in Ethiopia [33,38,46].

## Prevalence of knowledge, attitude, and practice on human and animal anthrax

The computation of the fixed-effects model revealed significant heterogeneity (p<0.001) among the included studies. Therefore, we pooled the effect size of knowledge on anthrax prevention and control using random-effects model analysis. As a result, the pooled level of anthrax knowledge was 51.25% (95% CI: 43.93, 58.58; $I^2$ = 98.01%) (Fig 2).

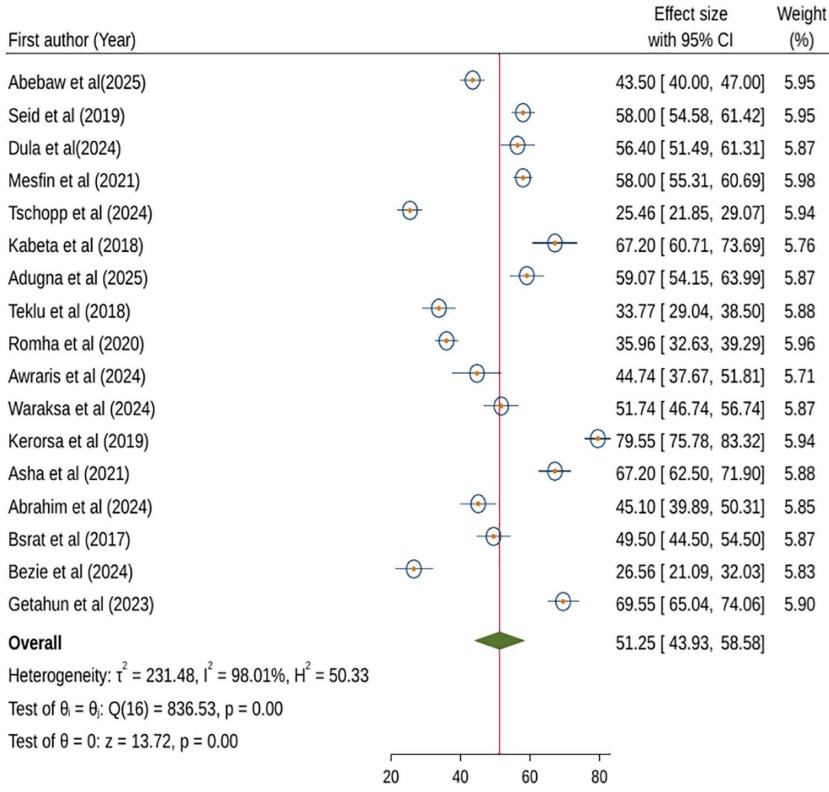

**Fig 2. Forest plot of pooled knowledge on anthrax prevention and control, Ethiopia, 2025.**

Similarly, we used 14 included studies to estimate the effect size of attitude towards anthrax prevention and control using a random-effects model. The result showed that 59.26% (95% CI: 50.43%, 68.08%; $I^2 = 98.33\%$; p < 0.001) was the pooled prevalence of attitude towards anthrax prevention and control (Fig 3).

Fifteen (15) studies were included in the meta-analysis to estimate the effect size using random-effects model analysis on participants' practice of anthrax preventive and control measures because fixed-effects model analysis showed significant heterogeneity (p < 0.001) across the included studies. The percentage of study participants in Ethiopia that practiced anthrax prevention and control strategies was 50.62% (95% CI: 42.95, 58.29; $I^2 = 97.93\%$; p < 0.001) (Fig 4).

## Subgroup analysis for KAP outcomes

Region, population type, and study year categories were used to do subgroup analyses for the knowledge, attitude, and practice outcomes. The Oromia region had the highest percentages for knowledge, attitude, and practice outcomes—56.89%, 64.12%, and 53.78%, respectively. According to population type, professionals had the highest percentage of knowledge (69.55%), livestock owners had the highest percentage of attitude (64.88%), and professionals had the highest percentage of practice (61.80%). Additionally, we divided the included studies into two groups: those conducted between 2016 and 2020 and those conducted between 2021 and 2025. Subgroup analysis using this category revealed notable variations in attitude and practice outcomes. The 2021–2025 study periods were found to have a higher proportion of attitude and practice outcomes, accounting for 64.57% and 54.56%, respectively (Table 2).

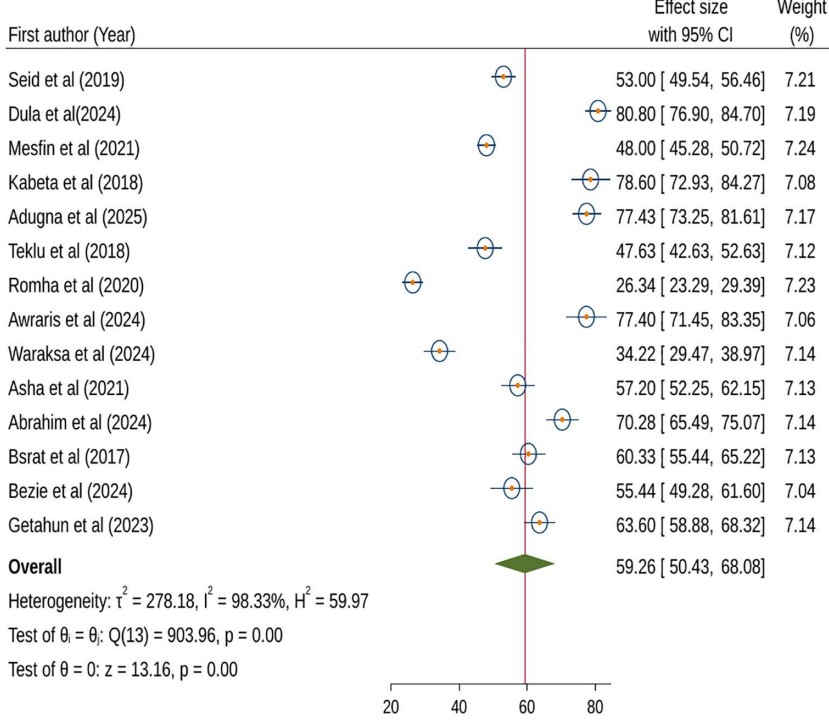

**Fig 3. Forest plot of attitude towards anthrax prevention and control in Ethiopia, 2025.**

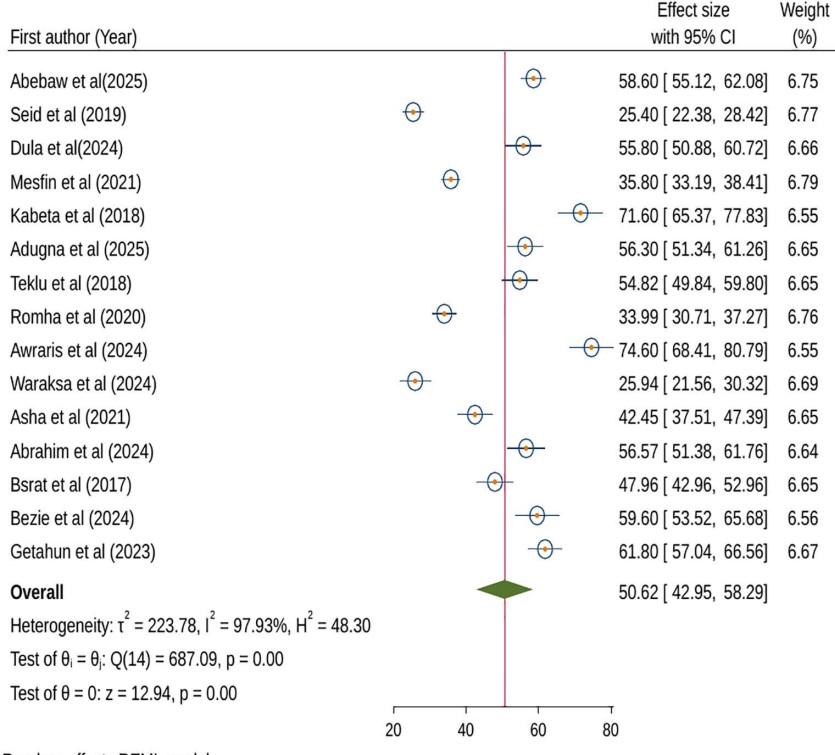

**Fig 4. Forest plot of pooled practice on anthrax prevention and control in Ethiopia, 2025.**

## Meta-regression analysis

With $I^2$ values of 98.01%, 98.33%, and 97.93%, respectively, the meta-analyses revealed notable high levels of heterogeneity among the included studies in the estimation of knowledge, attitude, and practice effect sizes. Therefore, using sample size and publication year as covariates, we performed meta-regression analysis to determine the potential sources of heterogeneity. The findings demonstrated that the presence of heterogeneity between studies for the estimate of knowledge and attitude outcomes was not significantly influenced by either sample size or publication year. However, the meta-regression analysis demonstrated that sample size had a statistically significant impact on the presence of study heterogeneity in the estimation of practice result (p = 0.031). This suggests that the observed heterogeneity between studies was influenced by differences in sample sizes. According to meta-regression, the pooled level of good practice regarding anthrax dropped by 2.9% for every 100 participants in the sample (β = -0.029, p = 0.031) (Table 3).

## Publication bias and sensitivity analysis

By computing the Egger's test and inspecting funnel plots, we checked whether publication biases were present in the effect size estimation of KAP outcomes. The results of the Egger's test for attitude (p = 0.13) and knowledge (p = 0.99) were statistically non-significant. Furthermore, symmetric inspection of funnel plots was performed for the estimate of both outcomes. These demonstrated that there was no publication bias. Nonetheless, both Egger's test and the asymmetric inspection of the funnel plot demonstrated the existence of publication bias with relation to the estimation of practice outcome. Significant asymmetry was found using funnel plot analysis, with two studies close to the midline, four on the left, and nine on the right (Fig 5). Significant publication bias was found, as demonstrated by Egger's test (p < 0.001). There

**Table 2. Subgroup analysis of KAP outcomes on anthrax in Ethiopia by regions, population type, and study year, 2025.**

| KAP Outcome | Subgroups | Categories | Number of studies | Pooled proportion (95% CI) | I² |
|---|---|---|---|---|---|
| Knowledge | Regions | Oromia | 9 | 56.89(47.78, 66.0) | 96.86% |
| | | Amhara | 5 | 46.3(34.92, 57.69) | 97.62% |
| | | Others* | 3 | 42.83(18.31, 67.4) | 99.19% |
| | Population | Livestock owners (Farmers) | 7 | 51.04(41.0, 61.08) | 97.45% |
| | | General community | 8 | 52.41(41.72, 63.1) | 97.82% |
| | | Pastoralists | 1 | 25.46(21.85, 29.1) | 0.0% |
| | | Professionals | 1 | 69.55(65.04, 74.1) | 0.0% |
| | Study years | 2016-2020 | 8 | 50.90(38.2, 63.6) | 98.88% |
| | | 2021-2025 | 9 | 51.59(42.87, 60.3) | 96.56% |
| Attitudes | Regions | Oromia | 8 | 64.12(52.75, 75.5) | 97.89% |
| | | Amhara | 4 | 58.30(45.6, 70.9) | 97.3% |
| | | Others* | 2 | 41.71(11.46, 71.9) | 99.08% |
| | Population | Livestock owners (Farmers) | 6 | 64.88(53.42, 76.3) | 97.68% |
| | | General community | 7 | 53.81(39.5, 68.14) | 98.69% |
| | | Professionals | 1 | 63.60(58.9, 68.32) | 0.0% |
| | Study years | 2016-2020 | 6 | 52.22(38.53, 65.9) | 98.71% |
| | | 2021-2025 | 8 | 64.57(53.8, 75.34) | 97.52% |
| Practice | Regions | Oromia | 8 | 53.78(44.65, 62.9) | 96.24% |
| | | Amhara | 5 | 50.67(33.28, 68.1) | 99.07% |
| | | Others* | 2 | 38.01(29.73, 46.3) | 87.19% |
| | Population | Livestock owners (farmers) | 7 | 54.41(41.04, 67.8) | 98.67% |
| | | General community | 7 | 45.29(36.5, 54.11) | 96.18% |
| | | Professionals | 1 | 61.80(57.04, 66.6) | 0.0% |
| | Study years | 2016-2020 | 6 | 44.77(31.5, 58.1) | 98.67% |
| | | 2021-2025 | 9 | 54.56(45.7, 63.42) | 96.69% |

Note: Others*=Afar/Somalia, Tigray, and SNNP regions.

**Table 3. Meta-regression analysis of factors influencing heterogeneity between studies, 2025.**

| KAP Outcome | Factors | Coefficients | Std.Err | P-value |
|---|---|---|---|---|
| Knowledge | Publication Year | -1.474031 | 1.50029 | 0.343 |
| | Sample Size | -.0023216 | .0148213 | 0.878 |
| Attitude | Publication Year | 1.021309 | 1.711052 | 0.563 |
| | Sample Size | -.0265073 | .015764 | 0.121 |
| Practice | Publication Year | .8618538 | 1.268949 | 0.51 |
| | Sample Size | -.0290356 | .0118669 | 0.031 |

may be small-study effects and selective publication of studies with larger estimates, as the asymmetry pattern suggests that smaller studies tended to report higher levels of good practice toward anthrax.

A trim-and-fill analysis was performed to take into account for the publication bias that was detected. While the Egger's regression test revealed considerable asymmetry ($p < 0.001$), no potentially missing research were found using the trim-and-fill method. It is possible that the asymmetry was not caused by missing studies or that there were not enough studies available for meaningful imputation, even in the face of statistical evidence of publication bias (Fig 6). After adjustment, the pooled estimate of good practice (50.62%) stayed the same.

 PLOS Neglected Tropical Diseases

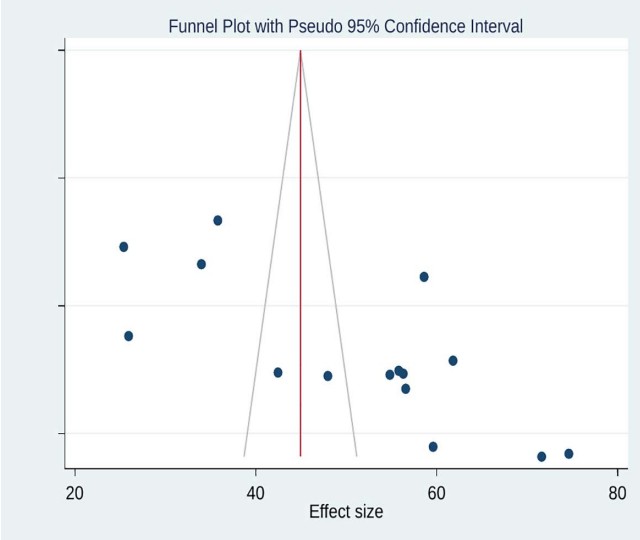

**Fig 5. Funnel plot of testing publication bias for 15 studies, practice, 2025.**

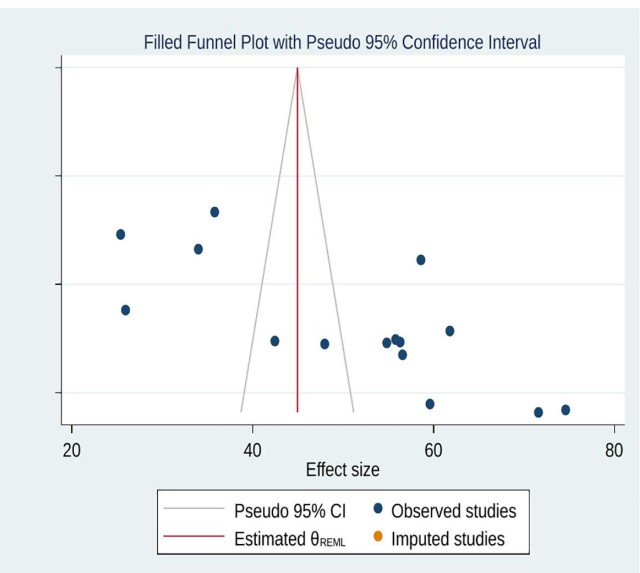

**Fig 6. Result of trim and filled analysis for adjusting publication bias of the 15 studies.**

## Sensitivity analysis

The impact of each included study in the pooled estimates of knowledge, attitude, and practice about anthrax was evaluated using sensitivity analysis. Upon progressively excluding each study, the pooled proportions of adequate knowledge, attitude, and practice were 49.47%-52.78%, 57.58%-61.81%, and 48.93%-52.44%, respectively. All pooled estimates remained statistically significant (p<0.001). The robustness of the findings is confirmed by the fact that no single study had a disproportionate impact on the overall pooled estimations.

## Discussion

This systematic review and meta-analysis synthesized data on risk factors and knowledge, attitude, and practice (KAP) levels related to anthrax prevention and control in Ethiopia. the study revealed that raw meat consumption, backyard slaughtering, cultural attitudes, co-habitation with animals, and inappropriate carcass disposal were common risk factors of anthrax infection. In meta-analysis, the pooled estimates of knowledge (51.25%), attitude (59.26%), and practice (50.62%) showed significant gaps in community readiness to prevent and control anthrax in Ethiopia.

Our synthesis revealed that in Ethiopia, shared housing with animals, raw meat intake, backyard slaughtering, and inappropriate carcass disposal were behavioural risk factors for anthrax infection. Concordant to our investigation, there is practice of consuming meat from suddenly died animals in Uganda [19] and Zambia [47], and slaughtering of sick animals in Bangladesh [48]. Global syntheses also demonstrated that outbreaks are frequent when individuals are led by economic factors to slaughter or consume animals that suddenly die, which supports our finding [15]. Social and health system gaps, such as inadequate veterinary care, the economic need of meat, cultural eating customs, and inadequate infrastructure for disposing of carcasses, explain why people frequently engage in risky behaviours even when they are aware of them [19].

The good knowledge of population about anthrax in Ethiopia was pooled to be 51.25%, according to our meta-analysis; only half of at risk populations have adequate understanding of anthrax prevention and control strategies. This finding is nearly similar to that of studies conducted in Kenya [49] and Bangladesh [50] that reported overall respondents' knowledge of anthrax to be 51% and 55.8%, respectively. However, it is lower when compared to that of studies conducted in Uganda (77%) [19] and Nigeria (58.6%) [51]. The disparities could be explained by differences in measurement, education, communication, and health system engagement, as well as differences in exposure to previous outbreaks. Subgroup analysis on knowledge outcome found that the Oromia region had the greatest pooled estimate of knowledge (56.89%), followed by Amhara (46.3%) and other regions such as Afar, Somalia, and Tigray (42.83%).,. These discrepancies most likely stem from regional variances in anthrax endemicity, livestock management practices, exposure to education, public health initiatives, and may be due to variation in the number of included studies. Moreover, Oromia's comparatively higher level of knowledge could be explained by the region's higher cattle population and more frequent anthrax outbreaks, which have raised awareness. The regional disparity indicates that national anthrax prevention measures need to be focused on specific areas to be effective.

The subgroup analysis based on study period (2016–2020 vs. 2021–2025) revealed that the year of study was not a major mediator of knowledge variance, as seen by the slight difference in pooled knowledge estimates (50.90% and 51.59%, respectively). This implies that, despite governmental initiatives One Health policy, understanding of anthrax in Ethiopia has stayed stable over the past ten years [52]. The absence of a significant temporal pattern in the Ethiopian data thus indicates to a crucial implementation gap: awareness and education initiatives are frequently reactive and outbreak-driven rather than being sustained in the routine provision of veterinary and public health services. The aforementioned highlights the necessity of on-going One Health communication and surveillance initiatives that can sustain and expand community awareness. Using population as subgroup category, good knowledge level was found comparable among livestock owners and general communities (51.04% vs 52.41%), while it was found largest (69.55%) among professionals and lowest (25.46%) among pastoralists. This variation might be contributed by unstable living style of pastoralists limiting them to be accessible to public awareness programs. Inclusion of only one study in this group of population might also be attributable to this variation.

The pooled estimate of attitude towards anthrax prevention and control in Ethiopia is 59.26% which is lower when compared to other studies conducted in Nigeria (79.9%) [51], in Zambia (80%) [47] and in China (75.8%) [53]. Variability among nations is probably due to a variety of factors, including recent outbreak history, individual anthrax experiences, the scope and content of public health education, sociocultural variances, and methodological and population variations.

Attitude showed regional differences in subgroup analysis, with Oromia having the highest positive attitudes (64.88%), Amhara coming in second (58.30%), and other regions having the lowest (41.71%). The lowest attitude score (41.71%) in other regions might be due to a lack of veterinary extension services, a lack of health education coverage, and a lack of community experience with anthrax outbreaks. This is consistent with studies that official health education and immunization programs are frequently less accessible in remote or pastoral areas, which results in lower awareness and unfavourable attitudes toward preventing zoonotic diseases [44]. The subgroup analysis based on population type revealed that the pooled attitude towards anthrax prevention and control was similar among livestock owners (64.88%) and health professionals (63.6%), while the general community showed a relatively lowest attitude score (53.81%). This pattern implies that professional involvement in animal and human health services or direct animal exposure may raise awareness and promote more positive attitudes regarding anthrax prevention. The zoonotic potential of the disease is frequently underestimated in communities with little animal interaction or no past outbreak history, which results in a lack of prevention behaviours [44]. Overall, this finding highlights the necessity of widespread community health education for the general population as well as farmers and professionals. All population groups can have more positive views about anthrax prevention if One Health approach that collaborate animal and human health messaging are strengthened.

Meta-analysis evidenced that only half (50.62%) of respondents follow the recommended anthrax preventive measures in Ethiopia. This figure is similar to studies from other developing nations where anthrax is still widespread. For instance, 52% of livestock producers in Bangladesh reported that they vaccinated and disposed of carcasses safely during the outbreak [54]. However, it is lower when compared to a study conducted in Kazakhstan where 70% of participants demonstrated practice of preventing anthrax [55]. The discrepancy might be attributable to differences in culture, health system context, population, and socio-economic conditions. The subgroup analysis using study years showed that good practice for anthrax prevention and control in Ethiopia enhanced from 44.77% (2016–2020) to 54.56% (2021–2025). This increasing tendency shows a slow but steady improvement in community involvement and commitment to preventive strategies over time. However, the 54.56% practice level is still below optimal, reflecting that almost half of the communities continue to practice risky behaviours like handling carcasses improperly or consuming raw meat. Overall, the substantial heterogeneity observed across studies likely reflects differences in population groups, KAP measurement approaches, and study years and settings; therefore, pooled estimates of KAP outcomes should be interpreted with caution and with greater emphasis on subgroup-specific findings.

In estimating the effect sizes of knowledge and attitude against anthrax prevention and control, our meta-analysis evidenced no publication bias. However, Egger's test and funnel plot inspection revealed the existence of publication bias in estimating pooled practice level, with an asymmetry (four studies laid on the left and nine on the right). We performed the trim-and-fill analysis to adjust; however, the result showed no change in the pooled estimate, indicating that the bias had no effect on the overall estimation. The robustness of findings was further checked by sensitivity analysis; no single study had impact on the overall estimates. Additionally, sample size was found to be a significant covariate in meta-regression analysis, indicating that variations in sample sizes might contribute to high degrees of heterogeneity across the included studies. Smaller studies might tend to report high prevalence of practice on anthrax prevention and control. This finding indicates the presence of potential small study effects, where studies with smaller sample sizes yield higher outcome causing existence of publication bias [56,57].

## One health implications

According KAP findings in this systematic review and meta-analysis, the communities in Ethiopia have suboptimal levels of knowledge, attitude, and practice towards anthrax prevention and control strategies. In light of these gaps, there is a stronger implication of implementing One Health policy that incorporates the sectors of environmental, animal, and human health [58]. Poor animal handling practices and inadequate awareness contribute to continuous animal-human anthrax transmission, particularly in rural residents who rely heavily on livestock production. Enhancing collaborative sector efforts,

public education, and joint surveillance are paramount for improving the prevention and control of anthrax outbreaks [59]. Hence, the observed gaps in KAP among specific population groups suggest the need for targeted, sector-specific interventions within a coordinated One Health framework, rather than uniform population-wide strategies [15].

## Strengths and limitations

Our systematic review and meta-analysis provided insights into a One Health strategy in terms of community members' knowledge, attitudes, and practices about anthrax prevention and control in Ethiopia. It wasn't, however, done without limitations. First, representativeness for the whole country was limited because the majority of the included studies were conducted in specific regions (Oromia and Amhara). Second, there was significant heterogeneity in the estimation of KAP outcomes among the included studies, which may have resulted from differences in sample size, population, and measurements. Third, unpublished or unfavourable result studies might not be accessible, which could cause publication bias in our estimation. Fourth, the interpretation of subgroup-specific findings should be emphasised as the inclusion of heterogeneous population groups may limit the interpretation of overall pooled estimates. Fifth, we were unable to identify factors associated with KAP outcomes because the included studies either did not provide KAP factors or reported them inconsistently. Finally, because no previous systematic reviews and meta-analyses had been conducted both nationally and globally, KAP results were not discussed to comparable findings.

## Conclusion and recommendations

According to this systematic review, living with animals, handling carcasses improperly, consuming raw meat, and slaughtering animals are all common risk factors for contracting anthrax in Ethiopia. The pooled levels of knowledge, attitude, and practice regarding anthrax prevention and control are suboptimal, according to a meta-analysis. In Ethiopia, half of the communities lack adequate knowledge and continue to use unsafe anthrax prevention and control measures. This could result in underreported or delayed cases, inadequate outbreak response, and on-going animal-to-human transmission, especially in places where anthrax is endemic. These results suggest that, from a One Health policy perspective, there is an urgent need to improve collaboration between the environmental, animal, and human health sectors through collaborated, multisectoral efforts. To improve public knowledge, safe practices, and behaviours related to anthrax prevention and control, it is critical to strengthen integrated surveillance systems, access veterinary extension services, and improve public health education. Integrating One Health policy into regional and national disease prevention programs can improve sustainable knowledge, practical and behavioural changes to reduce anthrax transmission and its burden in Ethiopia and other low- and middle-income countries.

## Supporting information

**S1 File. PRISMA 2020 checklist [22].**
(DOCX)

**S2 File. Searching strategy strings.**
(PDF)

## Acknowledgments

We would like to acknowledge all authors of the primary studies included in our systematic review and meta-analysis.

## Author contributions

**Conceptualization:** Ayenew Takele Alemu, Gashaw Molla Beza, Mahider Awoke Belay.

**Data curation:** Ayenew Takele Alemu, Wolde Melese Ayele, Atirsaw Assefa Melikamu.

**Formal analysis:** Ayenew Takele Alemu, Birtuedil Yibeltal Beyene, Friehiwot Molla, Abathun Temesgen, Mekuanint Taddele Tessema, Melaku Laikemariam, Almaw Genet Yeshiwas.

**Investigation:** Ayenew Takele Alemu, Mahider Awoke Belay, Kindie Bayih Geremew, Abathun Temesgen.

**Methodology:** Ayenew Takele Alemu, Wolde Melese Ayele, Gashaw Melkie Bayeh.

**Project administration:** Ayenew Takele Alemu, Gashaw Molla Beza.

**Software:** Wolde Melese Ayele, Almaw Genet Yeshiwas, Atirsaw Assefa Melikamu.

**Supervision:** Friehiwot Molla.

**Validation:** Ayenew Takele Alemu, Birtuedil Yibeltal Beyene, Gashaw Melkie Bayeh, Kindie Bayih Geremew.

**Visualization:** Ayenew Takele Alemu.

**Writing – original draft:** Ayenew Takele Alemu, Gashaw Molla Beza, Mahider Awoke Belay.

**Writing – review & editing:** Ayenew Takele Alemu, Mekuanint Taddele Tessema, Melaku Laikemariam.

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
