## [Decision Letter · Decision Letter 0]

27 Dec 2025

Response to Reviewers
Revised Manuscript with Track Changes
Manuscript

Shaden Kamhawi

co-Editor-in-Chief

Paul Brindley

co-Editor-in-Chief

**Additional Editor Comments (if provided):**
**Journal Requirements:**

**Reviewers' comments:**

**Key Review Criteria Required for Acceptance?**

**Methods**

-Are the objectives of the study clearly articulated with a clear testable hypothesis stated?

-Is the study design appropriate to address the stated objectives?

-Is the population clearly described and appropriate for the hypothesis being tested?

-Is the sample size sufficient to ensure adequate power to address the hypothesis being tested?

-Were correct statistical analysis used to support conclusions?

-Are there concerns about ethical or regulatory requirements being met?

Reviewer #1: -The general objective is stated, but no explicit or testable hypothesis is formulated, and the aims could be more clearly defined.

-A systematic review and meta-analysis is appropriate for synthesizing KAP evidence, although methodological reporting needs clarification.

-The population is described, but the included groups (community, livestock owners, professionals) are heterogeneous, which complicates interpretation.

-The total sample across studies is large

Reviewer #2: The authors conducted a systematic review and meta-analysis. The review protocol was registered in PROSPERO. The objective of the review could be improved. The search strategy could be improved. It is not clear and a justification is needed as to why the years for search strategy was not stated. Also only 17 studies were eligible from 2016 to 2025. Any explanation as to why no studies were eligible before 2016?

Reviewer #3: Yes

**Results**

-Does the analysis presented match the analysis plan?

-Are the results clearly and completely presented?

-Are the figures (Tables, Images) of sufficient quality for clarity?

Reviewer #1: - The main planned analyses (random-effects meta-analysis, subgroup analyses, meta-regression, and publication bias assessment) are performed, but some analytical steps—such as harmonizing KAP definitions—are not clearly aligned with the stated plan.

- The results are detailed and comprehensive, but high heterogeneity is insufficiently interpreted, and some sections (e.g., One Health findings, local disease names) are overly descriptive and not essential to the core results.

- The tables and forest plots are informative

Reviewer #2: Yes, the results match the analysis plan and the results are well presented and explained. Figures and tables are appropriate.

However, the abstract requires improvement to include the reference for the protocol registration, search strategy with key terms. Include inclusion and exclusion criteria.

Reviewer #3: Yes

**Conclusions**

-Are the conclusions supported by the data presented?

-Are the limitations of analysis clearly described?

-Do the authors discuss how these data can be helpful to advance our understanding of the topic under study?

-Is public health relevance addressed?

Reviewer #1: - The main conclusions align with the overall findings

- Some limitations are acknowledged, but major issues such as inconsistent KAP definitions, regional imbalance of studies, and extreme heterogeneity require stronger emphasis.

- The discussion highlights the need for One Health approaches and improved community awareness, though the link between findings and actionable insights could be more specific.

-The manuscript clearly outlines public health implications, particularly regarding anthrax transmission, outbreak preparedness, and One Health policy needs.

Reviewer #2: Conclusions presented reflect the results of the data analysis. Limitations presented well. Further limitation to explain why only a few studies were conducted in a few regions in Ethiopia.

General write up needs improvement in terms of grammar and some statements like line73-74; 88-89 lacking references. line 287, 384 with unclear statements.

Reviewer #3: Yes

**Editorial and Data Presentation Modifications?**

Reviewer #1: - Several sections—particularly the Introduction and Discussion—contain redundant descriptions of anthrax epidemiology and general background information. Streamlining these sections would improve focus.

- PRISMA guidelines require transparent reporting of the complete search strings for each database. These should be added as supplementary material.

- The reported search period (August–September 2025) and the inclusion of studies up to 2025 require verification and correction.

- Clarify reporting of grey literature : specify whether unpublished studies were identified, assessed, or excluded.

- Strengthen the Limitations section : mphasize the high heterogeneity, variability in outcome definitions, and regional imbalance of included studies.

Reviewer #2: Minor as indicated in the previous sections

Reviewer #3: No

**Summary and General Comments**

Reviewer #1: This manuscript addresses an important and timely public health issue in Ethiopia by synthesizing national-level evidence on community knowledge, attitudes, and practices (KAP) related to anthrax prevention and control. The topic is highly relevant to One Health policy implementation. The authors present the first pooled national estimates on KAP outcomes for anthrax, which represents a notable contribution to the existing literature.

Strengths of the study include:Clear public health and One Health relevance ; A comprehensive search across multiple databases ; Use of established frameworks (PRISMA, JBI, CoCoPop) ; Application of appropriate statistical approaches such as random-effects meta-analysis, subgroup analysis, publication bias assessment, and sensitivity testing ; Inclusion of detailed contextual information that highlights real-world risk factors and behaviors contributing to anthrax transmission.

However, several weaknesses limit the clarity and interpretability of the findings: Definitions of “good knowledge,” “positive attitude,” and “good practice” are not standardized across primary studies, leading to inconsistent outcome measurement ; The introduction and discussion contain redundancies and could be made more concise and focused : The implications for One Health policy, although noted, would benefit from more actionable and specific recommendations.

Overall, the manuscript is well-conceived and addresses a significant gap in the field. The revisions required are primarily editorial and related to improved transparency, clarification of methodology, and stronger discussion of limitations rather than new analyses or additional data collection.

Reviewer #2: Abstract requires improvement and clarity for search strategies should be provided. Include in the methods sections how different databases were search based on Boolean Operators. Check for grammer and improper statements

Reviewer #3: The authors reviewed and analyzed previous studies on public health issues related to anthrax infections in Ethiopia. These studies covered transmission routes and residents' understanding of infection control measures. While information covering the entire country is unavailable, no similar meta-analysis has been conducted to date. This report contains valuable information for developing anthrax control measures in Ethiopia.

PLOS authors have the option to publish the peer review history of their article (what does this mean? ). If published, this will include your full peer review and any attached files.

**Do you want your identity to be public for this peer review?** For information about this choice, including consent withdrawal, please see our Privacy Policy .

Reviewer #1: No

Reviewer #2: **Yes:** Lawrence Mugisha

Reviewer #3: No

**Figure resubmission:**

**Reproducibility:** To enhance the reproducibility of your results, we recommend that authors of applicable studies deposit laboratory protocols in protocols.io, where a protocol can be assigned its own identifier (DOI) such that it can be cited independently in the future. Additionally, PLOS ONE offers an option to publish peer-reviewed clinical study protocols. Read more information on sharing protocols at https://plos.org/protocols?utm_medium=editorial-email&utm_source=authorletters&utm_campaign=protocols

---

## [Editor Report · Decision Letter 1]

13 Jan 2026

Dear Mr Alemu,

We are pleased to inform you that your manuscript 'A Systematic Review and Meta-analysis of Knowledge, Attitudes, and Practices toward Anthrax Prevention and Control in Ethiopia: Implication for a One Health Policy' has been provisionally accepted for publication in PLOS Neglected Tropical Diseases.

Best regards,

Richard A. Bowen, DVM PhD

Academic Editor

Paul Brindley

Editor-in-Chief

Shaden Kamhawi

co-Editor-in-Chief

Paul Brindley

co-Editor-in-Chief

Thank you for your comprehensive editing of your manuscript based on reviewer comments. This will be a valuable contribution to the field of anthrax epidemiology.

---

## [Editor Report · Acceptance letter]

Dear Mr Alemu,

We are delighted to inform you that your manuscript, "A Systematic Review and Meta-analysis of Knowledge, Attitudes, and Practices toward Anthrax Prevention and Control in Ethiopia: Implication for a One Health Policy," has been formally accepted for publication in PLOS Neglected Tropical Diseases.

Best regards,

Shaden Kamhawi

co-Editor-in-Chief

Paul Brindley

co-Editor-in-Chief
